# Efficacy of acupuncture in refractory irritable bowel syndrome: study protocol for a randomised controlled trial

Jun Zhao [ID] ,[1] Min Chen,[2] Xin Wang,[3] Kun Ye,[4] Suhua Shi,[5] Huixia Li,[6] Jianfang Wang,[7] Xiaowei Chen,[7] Jinxia Ni,[8] Qingshuang Wei,[1] Yunzhou Shi,[4] Yu Hu,[1] Jingwen Sun,[1] Da Li,[8] Siyuan Liu,[1] Zhigang Li,[1] Hui Zheng [ID] ,[4] Shu-guang Yu[4]

JZ, MC and XW contributed equally.

For numbered affiliations see end of article.

**Correspondence to**
Dr Zhigang Li;
lizhigang620@126.com

## ABSTRACT

**Introduction** Irritable bowel syndrome (IBS) is a common chronic functional gastrointestinal disorder that presents with abdominal pain/discomfort and altered bowel patterns. IBS has multiple potential causes for which conventional medicines have had limited success, resulting in a significant number of patients who do not sensitively respond to pharmacotherapy for a period of 12 months and who develop a continuing symptom profile (described as refractory IBS) and seek help through (non) pharmacological treatments. The aim of this study is to investigate the efficacy and safety of acupuncture therapy for refractory IBS on the basis of conventional treatments.

**Methods and analysis** A total of 170 eligible patients who meet the Rome IV criteria for refractory IBS will be randomly allocated to receive acupuncture or sham acupuncture. Each patient will receive 12 sessions of acupuncture over 4 weeks and a 4-week follow-up. The primary outcome will be the IBS Symptom Severity Score. Secondary outcomes will include the proportion of participants experiencing adequate relief of global IBS symptoms, the weekly frequency of defecation, the stool properties assessed by the Bristol Grading Scale, the Work and Social Adjustment Scale, the IBS-Quality of Life score, and the Self-Rating Depression Scale and Self-Rating Anxiety Scale anxiety and depression scores. Outcome measures will be collected at baseline, 2 and 4 weeks of the intervention, and 6 and 8 weeks after the intervention. Categorical variables will be compared with Fisher's exact test or the Wilcoxon rank-sum test, and continuous variables will be compared using Student's t-test or the Wilcoxon rank-sum test.

**Ethics and dissemination** The entire project has been approved by the ethics committees of Beijing University of Chinese Medicine (2020BZYLL0507) and Sichuan Province Regional Institution for Conducting Research on Traditional Chinese Medicine (2020KL-025). The outcomes of the trial will be disseminated through peer-reviewed publications.

**Trial registration number** NCT04276961.

## Strengths and limitations of this study

► This is the first multicentre randomised controlled trial verifying the benefit of acupuncture added to usual treatments for refractory irritable bowel syndrome (IBS).

► A sham acupuncture group will be used to investigate the placebo effect of acupuncture.

► This trial meets the methodological demand of adequate concealment of randomised group allocations, outcome assessors and statisticians.

► Participants will be screened for eligible refractory IBS at baseline through three rigorous phases.

► A limitation of this trial is that since a rigorous study design was developed to identify which patients with IBS are 'refractory', we must fully consider the differences in the severity of symptoms, the duration of symptoms, the dietary interventions or advice provided, and even the conventional pharmaceutical treatments provided.

5%–6% in China,[2] 9.5%–9.8% in Asia[3] and 11% worldwide,[4] which indicates that IBS is a major public health issue. Owing to the complex pathogenesis of IBS, several pathogenic factors (genetic and environmental factors, visceral hypersensitivity, abnormal gut motility, postinfectious inflammatory mechanisms, gut microbiota changes, psychological morbidity, and physical and emotional factors) have been proposed as possible causes.[5] Hence, the study of patients with IBS is still a challenge.

Furthermore, in clinical practice, patients with IBS complain about various symptoms, such as changes in bowel habits, abdominal pain, diarrhoea or constipation, and meteorism, which considerably worsen their quality of life (QOL). The Rome IV criteria suggest that IBS treatment is selected on the basis of the symptom type (abdominal pain, diarrhoea, etc).[1] Simultaneously, patients are stratified into four subtypes: constipation-predominant

## INTRODUCTION

Irritable bowel syndrome (IBS) is the most common functional gastrointestinal disorder worldwide.[1] Epidemiological data show that the prevalence of IBS is approximately

IBS, diarrhoea-predominant IBS, mixed IBS and unclassified IBS. Many medications for treating IBS-related symptoms have been developed for various IBS subtypes, including antispasmodics, psychotropic agents, bulking agents, 5-HT (5-hydroxytryptamine) receptor antagonists, antibiotics, probiotics and so on. Although 5-HT3 antagonists, 5-HT4 agonists, antispasmodics and alosetron have been shown to be useful for the treatment of IBS,[6] regrettably, some drugs, such as guanylate cyclase-C agonists and loperamide, have been proven to be inadequate for relieving symptoms.[6 7] Similarly, some psychological therapies, such as hypnotherapy, relaxation therapy and cognitive–behavioural therapy, have been proven to be effective treatments for IBS.[8] Even when these available treatments are used, symptoms can persist for more than 12 months in some patients with IBS. These patients with IBS can be considered therapy resistant (refractory).[9 10] Therefore, novel treatment options for IBS, especially refractory IBS, are urgently needed.

Recently, there has been an increase in the use of non-pharmacological therapies, including acupuncture, which is widely accepted as an effective alternative treatment for gastrointestinal disorders in clinical practice.[11] According to reviews of the literature, acupuncture is not only satisfactory for IBS but may also be more effective than pharmacological therapies in alleviating the symptoms and QOL, with its effects lasting up to 12 weeks,[12] even sustaining benefits over a period of 12 months postrandomisation.[13 14] However, there are still limitations within previous studies, including small sample sizes, poor quality of the trials and lack of detailed assessments of symptom improvement, such as social activity scales and psychological questionnaires. Meanwhile, few studies have been designed for investigating the treatment of refractory IBS with acupuncture. Therefore, the main objective of this trial is to evaluate, on the basis of conventional treatments, the efficacy of acupuncture versus sham acupuncture for refractory IBS in terms of symptom management.

## METHODS/DESIGN

### Study design

This multicentre randomised controlled trial (RCT) will be conducted at six hospitals in China: Dongzhimen Hospital Affiliated to Beijing University of Chinese Medicine; Dongfang Hospital Affiliated to Beijing University of Chinese Medicine; Third Hospital Affiliated to Beijing University of Chinese Medicine; Beijing Hospital of Traditional Chinese Medicine Affiliated to Capital Medical University; Clinical Medicine College/Teaching Hospital Affiliated to Chengdu University of Traditional Chinese Medicine; and Third Hospital/Acupuncture and Tuina School Affiliated to Chengdu University of Chinese Medicine. The study has been approved by ethics committees and registered on www.clinicaltrials.gov on 19 February 2020. The study protocol will follow the principles of the Declaration of Helsinki and clinical research to Good

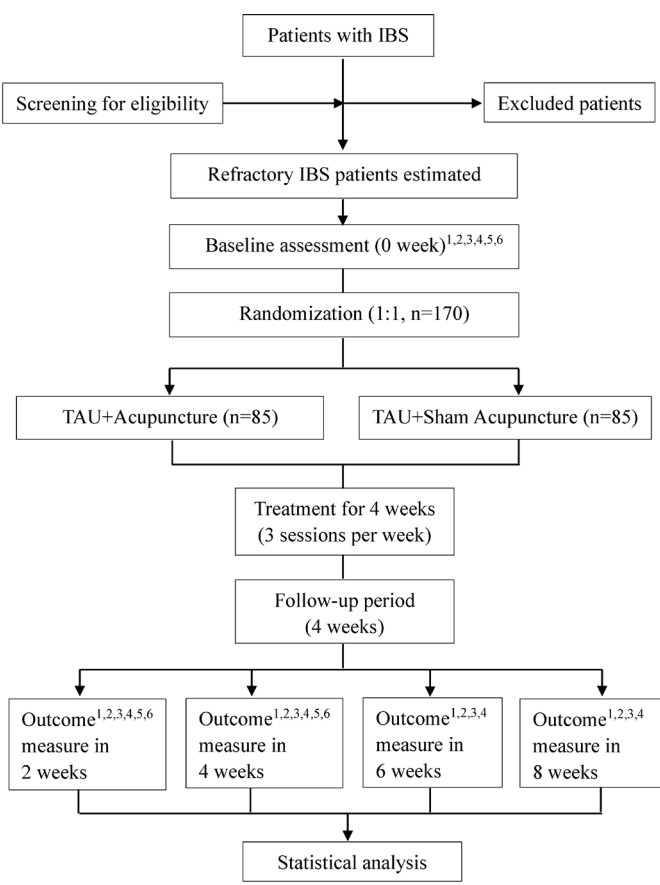

<sup>1</sup> is the IBS symptom severity score

<sup>2</sup> is the times of defecation weekly

<sup>3</sup> is the bristol stool form scale

<sup>4</sup> is the work and social adjustment score

<sup>5</sup> is the IBS-quality of life

<sup>6</sup> are the self-rating depression scale and self-rating anxiety scale anxiety and depression scores

TAU, treatment as usual

**Figure 1**  Trial flow diagram. IBS, irritable bowel syndrome.

Clinical Practice (GCP) guidelines, and will be reported based on SPIRIT (Standard Protocol Items: Recommendations for Interventional Trials) 2013 Checklist.[15] The checklist is provided in the online supplemental additional file 1.

Eligible patients will be randomly divided into two groups comprising participants who will receive 1 week of washout, 12 sessions over 4 weeks of treatment and 4 weeks of post-treatment follow-up. Outcomes will be assessed at baseline, during the treatment and at the end of the follow-up. A flow diagram of the trial is shown in figure 1.

### Patient recruitment

Step 1: Patients meeting the Rome IV IBS criteria (table 1) will be primarily recruited from the outpatient centres of the six participating hospitals. The mode of recruitment will also include advertisements through hospital social media sites (WeChat subscription), posters, Sojump, flyers about this trial, outpatient clinics and publicity at surrounding community service

**Table 1** Rome IV criteria: irritable bowel syndrome[45]

| Diagnostic time frame | Symptom description | Symptoms |
|---|---|---|
| ► ≥3 months of persistent symptoms with symptom onset at least 6 months before diagnosis<br>► ≥1 day/week | Abdominal pain | Recurrent association (two or more) of the following criteria:<br>► Related to defecation<br>► Associated with a change in the form of stool<br>► Associated with a change in the frequency of stool |

centres. Step 2: Patients (1) with IBS-related symptoms lasting for at least 12 months and (2) without a response to currently available IBS treatments, such as antibiotics or probiotics, dietary changes, antidepressants or psycho-therapies, will be included.[16–18] Clinicians will primarily select patients with suspected refractory IBS for further detailed assessment. Step 3: A clinical research assistant will identify eligible participants for this trial according to the following rigorous inclusion and exclusion criteria (box 1 and box 2). All patients will voluntarily agree to participate and will sign an informed consent document before randomisation.

### Randomisation/allocation

A block randomisation sequence for multiple centres will be generated by an independent professional statistician (Linkermed Tech) using Statistical Analysis System (SAS) software. All eligible participants will be randomly stratified into the acupuncture group or sham acupuncture group in a 1:1 ratio. Simultaneously, we plan to cooperate with a remote central web-based randomisation system. The clinical research assistant will be responsible for

---

**Box 1    Inclusion criteria**

**Items**
► Adults aged 18–70 years (either sex).
► Fulfilment of the Rome IV criteria for IBS.
► Patients with normal results of tests for occult blood in stool within the past month.
► Age above 50 years and normal results of colonoscopy within the previous year.
► Presence of symptoms for ≥12 months.
► Absence of response to a minimum of 6 weeks of dietary intervention or advice.
► Absence of response to an adequate dose of at least one conventional pharmacological agent tried for a minimum of 6 weeks.
► Signing of the written informed consent form.

IBS, irritable bowel syndrome.

---

**Box 2    Exclusion criteria**

**Items**
► Previous colonoscopy, meal barium fluoroscopy, abdominal ultra-sound and other examinations revealing severe intestinal organic lesions (including but not limited to ulcerative colitis, familial multi-ple intestinal polyps and colorectal cancers).
► The presence of one or more of the following warning symptoms: unexplained rectal bleeding, a positive faecal occult blood test re-sult; anaemia, abdominal mass, ascites, fever and emaciation.
► The presence of other severe medical conditions, such as cardio-vascular disease, endocrine disorders, hepatic dysfunction, renal diseases and cognitive disorders that can affect the outcome as-sessment results.
► Unconsciousness, inability to express subjective symptoms of dis-comfort and clearly diagnosed severe mental disorders.
► An unstable psychological state or accompanying psychological dis-orders (SDS>56).
► Present history of pregnancy or lactation.
► Acceptance of acupuncture treatment in the last 3 months.
► Difficulties in attending the trial, such as fear of acupuncture.

SDS, Self-Rating Depression Scale.

---

enrolling patients, screening patients, obtaining informed consent, requesting randomisation and collecting data.

### Blinding

Patients as well as research assistants, outcome assessors and statisticians will be blinded to the group assignment. However, due to the nature of acupuncture, the acupuncturists providing the interventions cannot be blinded. To minimise known sources of bias, the number/or location of the acupuncture points and manipulation of needles will be similar in the acupuncture and sham acupuncture groups, thereby optimising the blinding of subjects. Similarly, to avoid the subjective bias of researchers and subjects, the data analysis of the whole experiment will adopt a third-party blind evaluation. The participant's allocated intervention will not be revealed until the statistical analysis reports are finalised.

### Interventions

To ensure the safety and proper operation of acupuncture, all acupuncturists will participate in the standard operation process (SOP) of skin disinfection, location of acupoints and non-acupoints, fundamental puncture and needle manipulations before the trials. Treatments will be performed by licenced acupuncturists who have at least 3 years of experience in acupuncture. Acupuncture will be discontinued if the patients experience any adverse events (AEs) after acupuncture. The schedule of enrolment, interventions, assessments and participant visits is shown in figure 2.

### Treatment as usual

Treatment as usual (TAU) is defined as the continuation of current therapeutic agents prescribed by a general practitioner or gastroenterologist. All patients will continue TAU during the study period. Any changes

| | Study Period | | | | | | | | |
|---|---|---|---|---|---|---|---|---|---|
| | Enrollment | Allocation | Post-allocation | | | | | | Closeout |
| TIME POINT | w -1 | 0 | AFT | w 1 | w 2 | w 3 | w 4 | w 6 | w 8 |
| **ENROLLMENT:** | | | | | | | | | |
| *Eligibility screen* | × | | | | | | | | |
| *Informed consent* | × | | | | | | | | |
| *Randomization* | × | | | | | | | | |
| *Allocation* | | × | | | | | | | |
| **INTERVENTIONS:** | | | | | | | | | |
| *TAU+Acupuncture* | | | ◆━━━━━━━━━━◆ | | | | | | |
| *TAU+Sham Acupuncture* | | | ◆━━━━━━━━━━◆ | | | | | | |
| **ASSESSMENTS:** | | | | | | | | | |
| *IBS symptom severity score* | | × | | | × | | × | × | × |
| *Adequate relief of global IBS symptoms* | | | | × | × | × | × | | |
| *Bristol stool form scale* | | × | | | × | | × | × | × |
| *The frequency of defecation weekly* | | × | | | × | | × | × | × |
| *Work and social adjustment score* | | × | | | × | | × | × | × |
| *IBS-quality of life* | | × | | | × | | × | | |
| *The SDS and SAS scores* | | × | | | × | | × | | |
| *Blinding* | | | × | | | | × | | |
| *Safety evaluation* | | | | × | × | × | × | | |

**Figure 2** Schedule of enrolment, intervention and assessments of this study protocol. IBS, irritable bowel syndrome; SAS, Self-Rating Anxiety Scale; SDS, Self-Rating Depression Scale; TAU, treatment as usual.

in the medications of individual participants will be recorded on diary cards.

### Acupuncture group

Patients allocated to the acupuncture group with disposable and sterile acupuncture needles (length: 25–40 mm, diameter: 0.30 mm; Hwatuo, Suzhou, China) will receive treatment with acupuncture three times a week for 4 weeks for a total of 12 treatment sessions. Based on traditional Chinese medicine (TCM) theory and our clinical experience, the prespecified acupoints used here will include bilateral *Tianshu (ST25)*, *Shangjuxu (ST37)*, *Zusanli (ST36)* and *Neiguan (PC6)* (table 2 and figure 3). The exact location and depth of needling for each point will be determined based on the 2006 People's Republic of China National Standard: The Name and Location of

Acupoints (GB/T 12346-2006).[19] After insertion, manipulations of twirling, lifting and uniform reinforcing-reducing manipulation will be performed on all needles to elicit 'deqi'. The compositional sensation of 'deqi' includes numbness, soreness, distention, aching and heaviness, which are believed to be an essential component for acupuncture efficacy.[20] Then, the needles will be removed after a 30 min session with clean cotton balls to prevent bleeding.

### Sham acupuncture group

The participants in the sham acupuncture group will also receive the same treatment duration and frequency of sessions as the real acupuncture group. Similarly, sham acupuncture will be performed at predefined bilateral sites (non-acupoints 1–4) not corresponding

| Table 2 | Locations of acupoints in the acupuncture group | |
|---|---|---|
| **Acupoints** | | **Locations** |
| Tianshu (ST25) | Bilateral | On the same level of the umbilicus, and 2 cun* lateral to the anterior midline |
| Shangjuxu (ST37) | Bilateral | 6 cun directly below Dubi (ST35), and one finger-breadth lateral to the anterior border of the tibia |
| Zusanli (ST36) | Bilateral | 3 cun directly below Dubi (ST35), and one finger-breadth lateral to the anterior border of the tibia |
| Neiguan (PC6) | Bilateral | On the line joining Daling (PC7) and Quze (PC3), between the tendons of palmaris longus and flexor carpi radialis, 2 cun above the transverse crease of the wrist |

*1 cun (≈20 mm) is defined as the width of the interphalangeal joint of patient's thumb.

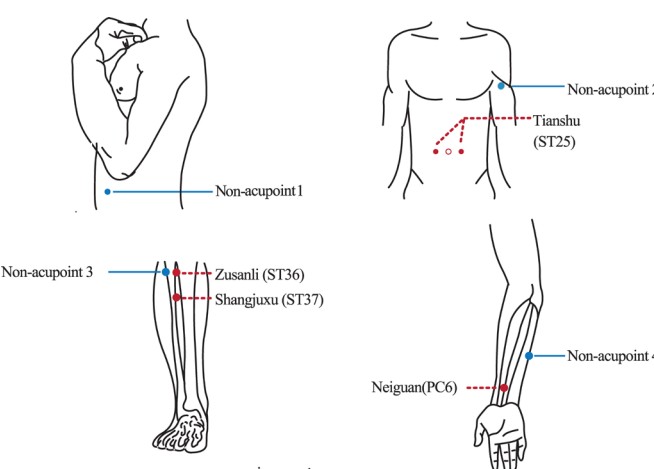

**Figure 3** Locations of acupoints and non-acupoints.

to acupuncture group traditional points or meridians, penetrating to the same depth but without needle manipulation for 'deqi' (table 3 and figure 3). Patients will be informed of the two types of acupuncture as follows: 'in this trial, different types of acupuncture will be compared, but another approach might not target your additional symptoms.' This language will be used to avoid deception while unifying the simulation conditions of the two groups as much as possible, mixing real and sham points, and this might enhance patient compliance.

## OUTCOMES
### Primary outcome
#### IBS Symptom Severity Score
According to previous assessment recommendations,[21] the IBS Symptom Severity Score (IBS-SSS) is a validated scoring system that relates to the severity of abdominal pain (IBS-SSS1), frequency of abdominal pain (IBS-SSS2), bloating (IBS-SSS3), satisfaction with bowel habits (IBS-SSS4) and disruption of QOL (IBS-SSS5), and will be used as the primary outcome of this trial. Participants will respond to each question on a 100-point Visual Analogue Scale. Each of the five questions generates a maximum score of 100 points, and the total is scored from 0 to 500, with higher scores indicating severe symptoms: normal (<75), mild IBS (75–175), moderate IBS (175–300) and severe IBS (>300). IBS-SSS will be assessed at baseline and 2, 4, 6 and 8 weeks after randomisation.

### Secondary outcomes
#### Adequate relief of global IBS symptoms
During our treatment period, patients will respond weekly to the question: 'In regard to all your symptoms of IBS, as compared with the way you felt before you added acupuncture therapy on the basis of conventional treatments, have you, in the past 7 days, had adequate relief of your IBS symptoms?' (yes/no). 'Yes' will be regarded as a positive response to treatment.[22]

#### Bristol Stool Form Scale
The Bristol Stool Form Scale classifies stool consistency into seven types encompassing complications from constipation to diarrhoea.[23] In this trial, participants will be asked to record the weekly defecation types and scores. The scores obtained from the Bristol scale will be assessed at baseline and weeks 2, 4, 6 and 8.

#### The weekly frequency of defecation
The frequency of bowel movements during the week before each visit will be assessed at baseline and 2, 4, 6 and 8 weeks after randomisation. A normal defecation is defined as a daily stool frequency ≤3 times and a stool consistency of type (like a sausage or snake; smooth and soft).[24]

#### Work and Social Adjustment Scale
The Work and Social Adjustment Scale (WSAS) will be assessed at baseline and weeks 2, 4, 6 and 8 using a five-domain scale with a total possible score of 40, with each item scoring from 0 (not affected) to 8 (severely affected). The scale is used to measure problems in functioning with work, home management, social activities, private leisure activities, and family and relationships.[25]

#### IBS-Quality of Life
The IBS-Quality of Life (IBS-QOL) questionnaire consists of 34 items; each item includes a five-point response scale and is a valid and reliable instrument to establish changes in health-related QOL of patients with IBS.[26] Patients themselves will be required to complete the questionnaire at baseline and weeks 2 and 4 after the treatment. Then, the total scores of each item will be calculated and analysed, with higher scores suggestive of better QOL.

| Table 3 | Locations of non-acupoints in the sham acupuncture group | |
|---|---|---|
| **Non-acupoints** | | **Locations** |
| Non-acupoint 1 | Bilateral | In the middle of Daheng (SP15) and Zhangmen (LR13) points (lateral aspect) |
| Non-acupoint 2 | Bilateral | In the medial arm on the anterior border of the insertion of the deltoid muscle at the junction of the deltoid and biceps muscles |
| Non-acupoint 3 | Bilateral | On the same level 1–2 cun* outboard lateral of the ST36 |
| Non-acupoint 4 | Bilateral | In the middle of the medial epicondyle of the humerus and the styloid process of ulna |

*1 cun (≈20 mm) is defined as the width of the interphalangeal joint of patient's thumb.

### Self-Rating Anxiety Scale and Self-Rating Depression Scale

The majority of patients with IBS have comorbid anxiety and/or depression, which are in turn strongly associated with symptom severity.[27 28] Hence, anxiety and depression symptoms will be assessed at baseline and weeks 2 and 4 using the Self-Rating Anxiety Scale (SAS) and Self-Rating Depression Scale (SDS). It is noteworthy that SDS >56 will be excluded before the first treatment because of an unstable psychological state or accompanying psychological disorders. According to statistical standards, the total score is multiplied by 1.25 and is then converted into a standardised score ranging from 25 to 100. For SAS, grades 1, 2 and 3 are related to scores of 53–62, 63–72 and >72, respectively.[29] For SDS, grades 1, 2 and 3 correspond to scores of 50–60, 61–70 and >70, respectively.[30] Higher scores indicate more severe anxiety and depression.

### Blinding assessment

To test the patient-blinding effects, all patients will be asked to guess whether they have received acupuncture or sham acupuncture at the end of the first treatment session and the 12th treatment session.

### Safety evaluation

In this trial, AEs will be monitored during the 4 weeks of treatment and the next 4 weeks of follow-up. Any adverse reactions that occurred during the study will be recorded. Common side effects of acupuncture or sham acupuncture include (but are not limited to) subcutaneous haematoma, local bleeding, skin bruising, needle site itching, continuous needle site pain, muscle spasm and dizziness. In the event of a serious AE, the necessary measures will be taken immediately for the safety of the subject. The time of occurrence, severity, duration, outcome and treatment-related status will be reported to the Medical Ethics Committee within 24 hours.

### Sample size and statistical analysis

Based on two sample size mean literature data speculation and previous clinical experience,[12 31] we anticipate the scores of two acupuncture groups on the continuation of current medications as follows: responders can improve refractory IBS on the IBS-SSS with scores of 30 and 15, respectively. The SD is set at 15, and the ratio between the acupuncture and sham acupuncture groups is 1:1. The sample size is calculated with a 20% type II error rate (80% power) and 5% type I error. The superiority trial will be adopted to minimise the difference between the two groups, and a 10-point is expected. In total, a sample size of 154 (77 patients in each group) is estimated. Allowing for 10% loss to follow-up, the sample size expands to 85 patients in each group.

All analyses will be performed using the intent-to-treat (ITT) and the per-protocol (PP) populations. ITT requires all participants to receive a baseline assessment of the primary outcome and at least one acupuncture session or one sham acupuncture administration. Missing values will be imputed through the multiple imputation method. The PP population is usually defined as patients who complete at least 80% of the treatment protocol without major protocol violations. The primary outcome will be assessed using analysis of covariance and adjusted for baseline total IBS-SSS. For other secondary outcomes, continuous variables will be analysed using Student's t-test or the Wilcoxon rank-sum test, and data will be represented as the mean±SD or median; categorical variables will be calculated using Fisher's exact test or the Wilcoxon rank-sum test, and data will be presented as frequencies (percentages). In terms of efficacy factors, other covariates, such as age, sex, disease duration, conventional pharmacological agent classification and so on, will be considered for further revision. The differences in blinding and AEs between the groups will be compared using the $\chi^2$ test. Moreover, we have decided to carry out a subgroup analysis based on patients with IBS subtypes, avoiding a heterogeneous patient population that may affect the efficacy results.

Statistical analysis of all data in this study will be performed by a specialised third-party statistician evaluation. SAS, V.8.2 statistical software (SAS Institute), will be used for data analysis. A two-sided $p<0.05$ will be considered statistically significant.

### Data management and quality control

To guarantee the objectivity of the data, individual participant data will not be disclosed outside. All researchers will receive training regarding data management. Data entry and management will be conducted using a database software clinical trial data management system by data administrators of the China Academy of Chinese Medical Science. An electronic case report form (CRF) will be designed to collect all clinical observation results of each participant before recruitment. The research assistant is responsible for verifying the accuracy and integrity of data to prevent any detection of errors, omissions or items requiring changes or clarification. Data lockup will be implemented by the data administrators on completion of the study. All paper files and electronic data will be kept for 5 years after publication and destroyed thereafter.

To guarantee the quality of this trial, a prespecified SOP, including recruitment and screening of participants, randomisation, laboratory detection results, intervention, details in filling the CRF, assessment of outcomes, data management and AEs of acupuncture, will be unified and trained through a multicentre internal study for consistency of results. Moreover, we will set up a Data and Safety Monitoring Board, an independent advisory group, to review and interpret data generated from the study, and a meeting will be held every 3 months to report the research progress. Each amendment of the protocol will conform to GCP principles and will maintain the ethical standards for RCTs.

### Patients and public involvement

Patients in this trial will not be involved in the design or conduct of the study or the outcome assessments.

However, we are planning to disseminate our research to the participants and the public, through means such as publicising our research on hospital social media sites (WeChat subscription) and various academic lectures.

## Ethics and dissemination

This study conforms to the principles of the Declaration of Helsinki and relevant ethical guidelines. The entire project has been approved by the ethics committees of Beijing University of Chinese Medicine (ID: 2020BZYLL0507) and Sichuan Province Regional Institution for Conducting Research on Traditional Chinese Medicine (ID: 2020KL-025). Written informed consent will be obtained from patients prior to enrolment in the study. The outcomes of the trial will be disseminated through peer-reviewed publications.

## DISCUSSION

IBS not only impacts patients' health-related QOL, resulting in a large economic burden on healthcare systems, but also impairs work productivity and individuals' ability to perform daily activities, increasing employers' indirect costs.[32] The currently available pharmacological treatments still have some limitations, resulting in many patients with refractory IBS seeking help through complementary medicine.[33]

However, when considering refractory IBS treatment, the symptoms should be highly addressed to maintain optimal results and enhance the QOL of individuals. Patients with IBS, especially refractory IBS, have more severe psychosomatic disorders.[34] Patients who are classified as having severe IBS agree with the statement: 'When my IBS is bad, I wish I was dead'.[35] Hence, another important treatment principle for refractory IBS is improving mental/emotional problems. From a TCM perspective, acupoint selection is a key factor in the success of acupuncture in treating diseases. According to previous studies and our previous reviews, the basic acupuncture points important for relieving diarrhoea symptoms are ST25, ST36 and ST37.[36 37] However, acupuncture treatment for psychosomatic diseases highlights the specific functions of acupoints. Clinical evidence has shown that PC6 can improve anxiety symptoms well.[38] Hence, the ST25, ST36, ST37 and PC6 points included in this trial are selected in accordance with not only the TCM treatment principles of 'syndrome differentiation' but also clinical practice guidelines in China. However, for these points, scientific evidence is sparse, and high-quality research is often lacking, leading to inconclusive results.[33] As such, a multicentre RCT with a large sample size is urgently needed to evaluate the efficacy of acupuncture in improving refractory IBS symptoms within the evidence-based medical framework.

In view of the complex psychological feelings of refractory patients, acupuncture treatment evaluations will be conducted in clinical settings in which usual treatment is routinely carried out. The many benefits of TAU include surmounting critical ethical issues (eg, withholding treatment)[39] while controlling for many common factors (eg, psychological concerns, dietary factors and medication changes) that can contribute to change. Moreover, because it is uncertain whether the long-term clinical beneficial effect of acupuncture on refractory IBS is sustained, another possible advantage of TAU is that research staff may establish good communication relationships with patients, thus avoiding potential dropouts. In this way, we can cautiously extrapolate the long-term effect of acupuncture.

Nevertheless, this study still has several limitations that should be noted. First, identifying which patients with IBS are 'refractory' is challenging because of the differences in the severity of symptoms, usual treatments administered and duration of symptoms.[40] On the basis of a literature review, patients with refractory IBS were defined as those who had an average daily pain rate ≥30 mm on the pain component scale of the IBS-SSS,[21] who failed to respond to usual medical treatment (education, dietary advice, antispasmodic agents, antidiarrhoeal medication, antidepressants, fibre-based medications, etc),[16–18 41 42] and who had IBS symptoms for 12 months or more.[9 10 42–44]

Second, to ensure a reliable sample of patients with refractory IBS, we plan to conduct two rounds of screening: In the first step, the clinicians will primarily select patients with suspected refractory IBS according to the above literature review items. Then, a clinical research assistant will recommend the eligible participants for this trial according to the following improved critical inclusion criteria: (1) there is an absence of response to an adequate dose of at least one conventional pharmacological agent tried for a minimum of 6 weeks; and (2) there is an absence of response to a minimum of 6 weeks of dietary intervention or advice. Moreover, we attach great importance to the potential risk of symptoms in refractory patients. Patients with one or more of the following warning symptoms will be excluded: unexplained rectal bleeding, a positive faecal occult blood test result; anaemia, abdominal mass, ascites, fever and emaciation. All participants will have to fill in the 'Filter table' in advance to guarantee that all inclusion criteria will be met.

We hope that at the end of this trial, the results will provide more reliable evidence and reveal the value of clinical acupoint selection guidelines in the treatment of refractory IBS with acupuncture.

## Trial status

This trial is currently in the preparation phase. No patient involved.

**Author affiliations**
[1]School of Acupuncture-Moxibustion and Tuina, Beijing University of Chinese Medicine, Beijing, China
[2]Chengdu University of Traditional Chinese Medicine Affiliated Hospital, Chengdu, Sichuan, China
[3]Capital Medical University Affiliated Beijing Hospital of Traditional Chinese Medicine, Beijing, China

[4]Chengdu University of Traditional Chinese Medicine, Chengdu, Sichuan, China
[5]Department of Rehabilitation, The Third Affiliated Hospital of Beijing University of Chinese Medicine, Beijing, China
[6]Department of Gastroenterology, Third Affiliated Hospital of Beijing University of Chinese Medicine, Beijing, China
[7]Department of Spleen, Stomach, Liver and Gallbladder Diseases, Dongfang Hospital Affiliated to Beijing University of Chinese Medicine, Beijing, China
[8]Department of Acupuncture and Moxibustion, Dongzhimen Hospital Affiliated to Beijing University of Chinese Medicine, Beijing, China

**Contributors** ZGL, HZ and S-GY conceived the study, they are co-corresponding authors. JZ, MC and XW are co-first authors. JZ, XW, KY, QW, YH, YS, JS, DL and SL are involved in the conception, design and critical revision for this study protocol. MC, HZ, XW, SS, HL, JW, XC and JN are responsible for recruiting subjects and assessment. ZGL and S-GY sought funding and ethical approval and helped with its implementation. HZ is responsible for revising the manuscript. All authors read and approved the final manuscript.

**Competing interests** None declared.

**Patient consent for publication** Not required.

**Provenance and peer review** Not commissioned; externally peer reviewed.

**ORCID iDs**
Jun Zhao http://orcid.org/0000-0002-3672-8963
Hui Zheng http://orcid.org/0000-0002-0494-1217

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
