## [Reviewer comments · BMJ Open]

ARTICLE DETAILS

TITLE (PROVISIONAL)	Efficacy of acupuncture in refractory irritable bowel syndrome: study protocol for a randomised controlled trial
AUTHORS	Zhao, Jun; Chen, Min; Wang, Xin; Ye, Kun; Shi, Suhua; Li, Huixia; Wang, Jianfang; Chen, Xiaowei; Ni, Jinxia; Wei, Qingshuang; Shi, Yunzhou; Hu, Yu; Sun, Jingwen; Li, Da; Liu, Siyuan; Li, Zhigang; Zheng, Hui; Yu, Shu-guang

VERSION 1 – REVIEW

REVIEWER	Sun, Jian-Hua Peking Union Medical College Hospital, Department of Critical Care Medicine
REVIEW RETURNED	19-Dec-2020

GENERAL COMMENTS	This is an interesting study. The authors evaluate investigate the efficacy and safety of acupuncture therapy for refractory IBS on the basis of conventional treatments. Whether the basic diseases of the included subjects are consistent. Please describe if there is any difference in routine medical care measures between the treatment group and the control group.
--

REVIEWER	Wu, Huangan Shanghai University of Traditional Chinese Medicine
REVIEW RETURNED	16-Feb-2021

GENERAL COMMENTS	This manuscript reports that the aim of this study is to investigate the efficacy and safety of acupuncture therapy for refractory IBS on the basis of conventional treatments. Although somewhat interesting, there are still some deficiencies need to be substantially improved, as detailed below. 1. The scientific quality as well as English writing/editing need to be substantially improved.2. Both of the acupuncture and sham acupuncture adopted the method to insert in the parts of the body. However, extensive research has shown that sham acupuncture has analgesic effect on patients with chronic pain. It has previously been observed that stimulation at many different parts, whether acupoints or not, would produce analgesic effect, so whether the design of the study could cause bias of the results? How do you solve this problem?3. Many randomized clinical trials exclude patients with a history of abdominal surgery. Why don't you? Do you think it is necessary to exclude it?
---

	4. Acupuncture group used the bilateral acupoints but sham acupuncture group don't indicate the acupoints is one side or two. Please clarify this.
--	--

VERSION 1 – AUTHOR RESPONSE

Response to reviewer 1:

Dr. Jian-Hua Sun, Peking Union Medical College Hospital

1. This is an interesting study. The authors evaluate investigate the efficacy and safety of acupuncture therapy for refractory IBS on the basis of conventional treatments. Whether the basic diseases of the included subjects are consistent.

Response: Dear reviewer, thank you for your comments.

According to the latest Chinese expert consensus of IBS and the 2016 Rome IV criteria^{1 2}, IBS is defined as “a functional bowel disorder in which recurrent abdominal pain is associated with defecation or a change in bowel habits. Disordered bowel habits are typically present, as are symptoms of abdominal bloating/distension occurring over at least 6 months and not less than 3 months.”

In this trial, participants will be screened for eligible refractory IBS at baseline through three rigorous phases. Step 1, patients meeting the Rome IV IBS criteria will be primarily recruited from the outpatient centres of the 6 participating hospitals. Step 2, patients 1) with IBS-related symptoms lasting for at least 12 months and 2) without a response to currently available IBS treatments. Clinicians will primarily select patients with suspected refractory IBS for further detailed assessment. Step 3, a clinical research assistant will identify eligible participants for this trial according to the following rigorous inclusion and exclusion criteria.

In addition, the case selection scale will be used to exclude patients with underlying basic diseases before the trial: 1) Previous colonoscopy, meal barium fluoroscopy, abdominal ultrasound and other examinations revealing severe intestinal organic lesions (including but not limited to ulcerative colitis, familial multiple intestinal polyps, and colorectal cancers); 2) The presence of one or more of the following warning symptoms: unexplained rectal bleeding, a positive faecal occult blood test result; anaemia, abdominal mass, ascites, fever, and emaciation; 3) The presence of other severe medical conditions, such as cardiovascular disease, endocrine disorders, hepatic dysfunction, renal diseases, and cognitive disorders that can affect the outcome assessment results.

With a passion for scientific study, we will try our best to ensure that the basic symptoms of the included subjects are consistent to verify the efficacy of acupuncture.

References

- 1 Chinese expert consensus of irritable bowel syndrome in 2020. *Chin J Dig* 2020;40:803-818.
- 2 Mearin F, Lacy BE, Chang L, *et al.* Bowel disorders. *Gastroenterology* 2016;S0016-5085:00222-5.

2. Please describe if there is any difference in routine medical care measures between the treatment group and the control group.

Response: Dear reviewer, thank you for your comments.

Due to IBS has multiple potential causes for which conventional medicines have had limited success, resulting in a significant number of patients who do not sensitively respond to pharmacotherapy for a period of 12 months and who develop a continuing symptom profile (described as refractory IBS) and seek help through (non)pharmacological treatments. According to reviews of the literature, acupuncture was not only satisfactory for IBS, but also may be more effective than pharmacological therapies in alleviating the symptoms and quality of life³, with its effects lasting up to 12 weeks, even sustaining benefits over a period of 12 months post-randomisation. However, rare studies have been designed for refractory IBS with acupuncture. The aim of this study is to investigate the efficacy and safety of acupuncture therapy for refractory IBS on the basis of conventional treatments. All participants will continue Treatment as usual (TAU) during the study period.

TAU is defined as the continuation of current medications prescribed by a general practitioner or gastroenterologist⁴. According to the TAU guidelines: 1) the first step consists of lifestyle modification and gut-targeted pharmacotherapy; 2) non-responders to the first step intervention proceed to the second step, which includes psychopharmacological agents in addition to the first step interventions. In this study, to ensure a reliable sample of refractory IBS patients, two inclusion criteria should be met: 1) there is an absence of response to a minimum of 6 weeks of lifestyle modification (dietary intervention or advice); 2) there is an absence of response to an adequate dose of at least one conventional pharmacological agent tried for a minimum of 6 weeks. We will ask all participants not to change their TAU as much as possible during the study period. Moreover, we will collect and record information about any changes in IBS treatments/management during the study. Any changes in medications of individual participant will be recorded by diary card.

In general, if patients met the inclusion criteria, there is no significant difference in patients (described as refractory IBS) who are not sensitively respond to lifestyle modification and/or pharmacotherapy at baseline (before enrollment). That is, there is no difference in routine medical care measures between the treatment group and the control group.

References

- 3 Pei L, Geng H, Guo J, *et al.* Effect of acupuncture in patients with irritable bowel syndrome: A randomized controlled trial. *Mayo Clin Proc* 2020;S0025-6196:30151-8.
- 4 Fukudo S, Kaneko H, Akiho H, Inamori M, Endo Y, Okumura T, *et al.* Evidence-based clinical practice guidelines for irritable bowel syndrome. *J Gastroenterol* 2015;50:11-30.

Response to reviewer 2:

Prof. Huangan Wu, Shanghai University of Traditional Chinese Medicine

1. The scientific quality as well as English writing/editing need to be substantially improved.

Response: Dear reviewer, thank you for your comments.

According to your comments, we checked through the English writing errors carefully, and invited professionals to help us with the revision.

2. Both of the acupuncture and sham acupuncture adopted the method to insert in the parts of the body. However, extensive research has shown that sham acupuncture has analgesic effect on patients with chronic pain. It has previously been observed that stimulation at many different parts, whether acupoints or not, would produce analgesic effect, so whether the design of the study could cause bias of the results? How do you solve this problem?

Response: Dear reviewer, thank you for your comments.

Actually, we quite agree with your point of view. Extensive research has shown that sham acupuncture has analgesic effect on patients with chronic pain. It has previously been observed that stimulation at many different parts, whether acupoints or not, would produce analgesic effect. With a passion for scientific study, I was involved in another study previously. Even if the blind method was achieved, the non-acupoint group was indeed effective. However, the benefit of acupuncture relies on acupoint specificity.

According to reviews of the literature and former experience, our design is as follows:

First, from a TCM perspective, acupoint selection is a key factor in the success of acupuncture in treating diseases. In our trial, the four non-acupoints are far from the traditional points and/or meridians. Non-acupoint 1: in the middle of Daheng (SP15) and Zhangmen (LR13) points (lateral aspect); Non-acupoint 2: in the medial arm on the anterior border of the insertion of the deltoid muscle at the junction of the deltoid and biceps muscles; Non-acupoint 3: on the same level 1-2 cun

outboard lateral of the ST36; Non-acupoint 4: in the middle of the medial epicondyle of the humerus and the styloid process of ulna.

Second, “deqi”, the acupuncture needling sensation, has been considered as an important component of acupuncture⁵. Analgesia manifests only when “deqi” occurs and acupuncture with “deqi” is found to have the best response⁶. In this trial, the participants in the sham acupuncture group will also receive the same treatment duration and frequency of sessions as the real acupuncture group, but without needle manipulation for “deqi”. According to the results of a previous questionnaire survey, the reliability of acupuncturists’ deqi sensation ranks as sinking>tightening>astringent. The reliability of patients’ deqi sensations ranks as sourness>numbness>distention> heaviness>pain⁷. Therefore, the deqi scale will be performed at the end of treatment to avoid bias of the results.

Third, this trial is a multicenter, randomized, controlled, superiority trial. Our other purpose is to try to answer the superiority of the acupuncture group. The latest research about “The challenges of evaluating specific and nonspecific effects in clinical trials of acupuncture for chronic pain” shows that acupuncture has statistically significantly better effects than sham acupuncture (effect sizes 0.16-0.19 [small]). When compared with usual care controls, effect sizes are larger (0.44-0.63 [moderate]). True acupuncture compared with usual care has an effect size of around 0.5, of which 60% is ascribed to nonspecific context effects plus sham, and the remaining 40% to the specific benefit of true acupuncture⁸. Similarly, Ma et al., indicated that acupuncture was effective in the treatment of functional dyspepsia, and was superior to non-acupoint puncture⁹. The benefit of acupuncture relies on acupoint specificity. According to previous studies and our previous reviews, the basic acupuncture points important for relieving diarrhoea symptoms are Tianshu (ST25), Zusanli (ST36) and Shangjuxu (ST37). However, acupuncture treatment for psychosomatic diseases highlights the specific functions of acupoints. Clinical evidence has shown that Neiguan (PC6) can improve anxiety symptoms well. Hence, the ST25, ST36, ST37 and PC6 points included in this trial are selected in accordance with not only the TCM treatment principles of “syndrome differentiation” but also clinical practice guidelines in China.

Moreover, to better understand the benefit of acupuncture, more comparative effectiveness research is needed ideally with pragmatic trial designs reflecting the real-world context. What is needed is to identify the conditions that acupuncture, as it is routinely delivered, might provide important clinical benefits for patients. According to reviews of the literature, acupuncture was not only satisfactory for IBS, but also may be more effective than pharmacological therapies in alleviating the symptoms and quality of life. As such, the main objective of this trial is to evaluate, on the basis of conventional treatments, the efficacy of acupuncture versus sham acupuncture for refractory IBS in terms of symptom management. We hope the results will provide more reliable evidence and reveal the value of clinical acupoint selection guidelines in the treatment of refractory IBS with acupuncture.

References

- 5 Zhao ZQ. Neural mechanism underlying acupuncture analgesia. *Prog Neurobiol* 2008;85:355-375.
- 6 Yang ES, Li PW, Nilius B, Li G. Ancient Chinese medicine and mechanistic evidence of acupuncture physiology. *Pflugers Arch* 2011;462:645-653
- 7 Ren YL, Guo TP, Du HB, Zheng HB, Ma TT, Fang L, et al. A survey of the practice and perspectives of Chinese acupuncturists on deqi. *Evid Based Complement Alternat Med* 2015;2015:684708.
- 8 MacPherson H, Charlesworth K. The Challenges of Evaluating Specific and Nonspecific Effects in Clinical Trials of Acupuncture for Chronic Pain. *Med Acupunct* 2020;32:385-387.
- 9 Ma TT, Yu SY, Li Y, et al. Randomised clinical trial: an assessment of acupuncture on specific meridian or specific acupoint vs. sham acupuncture for treating functional dyspepsia. *Aliment Pharmacol Ther* 2012;35:552-561.

3. Many randomized clinical trials exclude patients with a history of abdominal surgery. Why don't you? Do you think it is necessary to exclude it?

Response: Dear reviewer, thank you for your comments.

According to reviews of the literature, many randomized clinical trials exclude patients with a history of abdominal and/or rectal surgery. Actually, we previously quite agree with your point of view. It is necessary to exclude abdominal and/or rectal surgery. IBS as a common chronic functional gastrointestinal disorder which is characterized by symptoms of abdominal pain and/or discomfort with altered bowel habits (frequency of stool or form of stool), in the absence of an organic or structural cause, and symptoms onset should occur at least 6 months before diagnosis.

When we formulated the inclusion and exclusion criteria, we fully considered the suggestions of gastroenterologists, they believe that there are many kinds of digestive diseases, while IBS is mainly diagnosed by symptoms, in the absence of an organic or structural cause. Hence, patients with organic or structural causes should be excluded, but not limited to a history of abdominal and/or rectal surgery. Simultaneously, they also suggested that we revise the criteria for previously restricted diseases and change the words to "include but not limited to", that is, 1) Previous colonoscopy, meal barium fluoroscopy, abdominal ultrasound and other examinations revealing severe intestinal organic lesions (including but not limited to ulcerative colitis, familial multiple intestinal polyps, and colorectal cancers) should be excluded.

In other words, it is necessary to exclude abdominal and/or rectal surgery. If patients with a history of abdominal and/or rectal surgery, according to the first exclusion criterion, we will exclude them.

4. Acupuncture group used the bilateral acupoints but sham acupuncture group don't indicate the acupoints is one side or two. Please clarify this.

Response: Dear reviewer, thank you for your comments.

According to your comments, we have added relevant parameters of sham acupuncture group in the manuscript and tables. In this revised version, the content is marked in red.

Sham acupuncture will be performed at predefined bilateral sites (nonacupoints 1-4) not corresponding to acupuncture group traditional points or meridians, penetrating to the same depth but without needle manipulation for “deqi”. As shown in Table 5.

Table 5 Locations of non-acupoints in sham acupuncture group

Non-acupoints		Locations
Non-acupoint 1	Bilateral	In the middle of Daheng (SP15) and Zhangmen (LR13) points (lateral aspect)
Non-acupoint 2	Bilateral	In the medial arm on the anterior border of the insertion of the deltoid muscle at the junction of the deltoid and biceps muscles
Non-acupoint 3	Bilateral	On the same level 1-2 cun outboard lateral of the ST36
Non-acupoint 4	Bilateral	In the middle of the medial epicondyle of the humerus and the styloid process of ulna

*1 cun (≈ 20 mm) is defined as the width of the interphalangeal joint of patient's thumb